# MicroRNAs Expression in Response to rhNGF in Epithelial Corneal Cells: Focus on Neurotrophin Signaling Pathway

**DOI:** 10.3390/ijms23073597

**Published:** 2022-03-25

**Authors:** Chiara Compagnoni, Veronica Zelli, Andrea Bianchi, Antinisca Di Marco, Roberta Capelli, Davide Vecchiotti, Laura Brandolini, Anna Maria Cimini, Francesca Zazzeroni, Marcello Allegretti, Edoardo Alesse, Alessandra Tessitore

**Affiliations:** 1Department of Biotechnological and Applied Clinical Sciences, University of L’Aquila, Via Vetoio, 67100 L’Aquila, Italy; chiara.compagnoni@univaq.it (C.C.); veronica.zelli@univaq.it (V.Z.); roberta.capelli@graduate.univaq.it (R.C.); davide.vecchiotti@univaq.it (D.V.); francesca.zazzeroni@univaq.it (F.Z.); edoardo.alesse@univaq.it (E.A.); 2Center for Molecular Diagnostics and Advanced Therapies, University of L’Aquila, Via Petrini, 67100 L’Aquila, Italy; 3Department of Information Engineering, Computer Science and Mathematics, University of L’Aquila, Via Vetoio, 67100 L’Aquila, Italy; andrea.bianchi@graduate.univaq.it (A.B.); antinisca.dimarco@univaq.it (A.D.M.); 4Dompé Farmaceutici Spa, via Campo di Pile, 1, 67100 L’Aquila, Italy; laura.brandolini@dompe.com (L.B.); marcello.allegretti@dompe.com (M.A.); 5Department of Life, Health and Environmental Sciences, University of L’Aquila, P.zza S. Tommasi, 67100 L’Aquila, Italy; annamaria.cimini@univaq.it

**Keywords:** recombinant human NGF (rhNGF), microRNA, neurotrophin signaling pathway, corneal diseases, biomarkers

## Abstract

Purpose. Nerve growth factor efficacy was demonstrated for corneal lesions treatment, and recombinant human NGF (rhNGF) was approved for neurotrophic keratitis therapy. However, NGF-induced molecular responses in cornea are still largely unknown. We analyzed microRNAs expression in human epithelial corneal cells after time-dependent rhNGF treatment. Methods. Nearly 700 microRNAs were analyzed by qRT-PCR. MicroRNAs showing significant expression differences were examined by DIANA-miRpath v.3.0 to identify target genes and pathways. Immunoblots were performed to preliminarily assess the strength of the in silico results. Results. Twenty-one microRNAs (miR-26a-1-3p, miR-30d-3p, miR-27b-5p, miR-146a-5p, miR-362-5p, mir-550a-5p, mir-34a-3p, mir-1227-3p, mir-27a-5p, mir-222-5p, mir-151a-5p, miR-449a, let7c-5p, miR-337-5p, mir-29b-3p, miR-200b-3p, miR-141-3p, miR-671-3p, miR-324-5p, mir-411-3p, and mir-425-3p) were significantly regulated in response to rhNGF. In silico analysis evidenced interesting target genes and pathways, including that of neurotrophin, when analyzed in depth. Almost 80 unique target genes (e.g., *PI3K, AKT, MAPK, KRAS, BRAF, RhoA, Cdc42, Rac1, Bax, Bcl2, FasL*) were identified as being among those most involved in neurotrophin signaling and in controlling cell proliferation, growth, and apoptosis. AKT and RhoA immunoblots demonstrated congruence with microRNA expression, providing preliminary validation of in silico data. Conclusions. MicroRNA levels in response to rhNGF were for the first time analyzed in corneal cells. Novel insights about microRNAs, target genes, pathways modulation, and possible biological responses were provided. Importantly, given the putative role of microRNAs as biomarkers or therapeutic targets, our results make available data which might be potentially exploitable for clinical applications.

## 1. Introduction

Nerve growth factor (NGF) is a member of the neurotrophin family that includes also brain-derived neurotrophic factor (BDNF) and neurotrophins 3 (NT-3), 4/5 (NT-4/5), and 6 (NT-6). It plays a role by involving two classes of trans-membrane receptors: the high-affinity tropomyosin tyrosine kinase receptor (TrkA) and the low-affinity p75^NTR^ [1]. When bound to NGF, TrkA activates several pathways, such as the Ras-mitogen-activated protein kinase (MAPK), the extracellular signal-regulated kinase (ERK), the phospholipase C gamma (PLC-γ), and the phosphatidylinositol 3 kinase (PI3K). On the other hand, p75^NTR^ activates c-Jun kinase and NF-kB signaling, generates ceramide, and, in absence of co-expressed TrkA, leads cells toward apoptosis [2]. The biological role of NGF not only is crucial for central and peripheral nervous system function and maintenance but also is relevant for non-neuronal cells, such as immune-hematopoietic cells or epithelia. Besides several neuropathies, some studies [3,4,5,6,7,8] showed NGF clinical application for skin trauma and diseases (e.g., diabetic, pressure, and vasculitic ulcers), or also for ulcers of the eye’s anterior segment [8,9,10,11,12,13]. The molecule rhNFG (Cenegermin^®^), structurally identical to the human NGF, has been EMA (European Medicines Agency) and FDA (Food and Drug Administration) approved as first-in-class treatment for neurotrophic keratitis (NK), a rare degenerative corneal disease due to corneal trigeminal innervation defects causing spontaneous injuries and wounds [14,15,16]. The drug, administered as eye drops, acts directly on corneal epithelium by stimulating cell survival and growth. Furthermore, it promotes production of tears by lacrimal glands, thus providing lubrication and natural protection from pathogens and injury, and supports corneal innervation, usually lost in neurotrophic keratitis. Some side effects, such as hyperemia, ocular and periocular pain, and photophobia, are in general well tolerated by patients. Robust evidence highlights the possibility not only of treating NK with NGF but also possibly treating other corneal pathological conditions, due to the NGF capability of maintaining the homeostasis of the cornea in in vitro, ex vivo, and in vivo models [13,17]. In this regard, a recent phase-IIa study described the safety and efficacy of NGF for improving symptoms and signs of dry eye disease (DED) [18]. NGF and its TrkA receptor are both expressed in the anterior segment of the eye; in addition, NGF is released in the humor aqueous and tears and is produced by conjunctival cells (epithelial, goblet, immune cells, and fibroblasts) [12,13]. To date, the biomolecular response induced by NGF at the corneal level is still unknown. MicroRNAs (miRNAs) are short, non-coding RNA molecules able to post-transcriptionally regulate gene expression. They are transcribed as pri-miRNAs in the nucleus by RNA polymerase II. Pri-miRNAs are subsequently cleaved by Drosha Rnase III associated with the microprocessor complex subunit DGCR8 (DiGeorge critical region 8), thus generating hairpin pre-miRNA precursors that are moved to the cytoplasm by exportin 5. There, pre-miRNAs are cleaved again by Dicer, generating the mature double-stranded miRNAs that associate with a member of Argonaute family protein to produce the miRNA-induced silencing complex (miRISC), able to induce translation inhibition or mRNA instability and degradation [19]. MiRNAs are involved in maintaining cell homeostasis and fine-tuning fundamental biological processes, such as cell growth, proliferation, apoptosis, and metabolism. Dysregulated expression of miRNAs is observed and strictly linked to relevant human diseases (e.g., viral, immune, neurodegenerative diseases, cancer), thus indicating their relevant role in physio-pathological processes [20,21]. MiRNAs are found not only in cells but also in body fluids, such as blood, saliva, urine, and tears, where they can be released by active (i.e., secretion by exosomes or vesicles) or passive (i.e., cell death) mechanisms. MiRNAs in body fluids are stable and resistant to endogenous Rnase activity, due to the formation of macromolecular complexes (e.g., with Ago2 or HDL) or to exosome encapsulation [22,23,24,25,26]: for this reason, they are considered potentially suitable biomarkers or even therapeutic targets.

In this work, we analyzed miRNA profiling in a human corneal cell line (HCEpiCs) to detect expression levels in response to rhNGF. Among the differentially expressed miRNAs, we identified several miRNAs that were significantly modulated, unveiling interesting target genes and pathways.

## 2. Results

### 2.1. MiRNA Expression in HCEpiC Cells in Response to rhNGF

Expression levels of approximately 700 human miRNAs were evaluated in HCEpiCs after rhNGF treatment for 30 min and 12 and 48 h. Cell response to rhNGF was assessed by phosho-ERK1/2 induction and detection by Western blotting (Appendix A). Dysregulated miRNAs, identified by comparison with samples from different NGF time points with unstimulated control cells, are represented by volcano plots in Figure 1.

Among them, a total of 21 significant microRNAs were differentially expressed and reported above the represented *p* = 0.05 threshold (horizontal line) in at least one of the comparisons. As shown, most of microRNAs were downregulated after 12 and 48 h with respect to unstimulated cells, whereas upregulation was observed for several miRNAs in cells treated for 48 h, compared with 12 h. Global dynamic expression levels of the above-mentioned miRNAs are reported in Table 1. By comparing each NGF-treated sample with untreated cells, no relevant expression regulation was observed after 30 min. Conversely, miR-362-5p was significantly hypo-expressed after 12 h; miR-26a-1-3p and miR-146a-5p after 12 and 48 h; whereas eight miRNAs (miR-30d-3p, miR-27b-5p, miR-550a-5p, miR-34a-3p, miR-1227-3p, miR-27a-5p, miR-222-5p, miR-151a-5p) were significantly hypo-expressed after 48 h. On the other hand, by considering the sequential expression of miRNAs, based on time progression of treatments, miR-146a-5p, miR-449a, let7c-5p, miR-337-5p, miR-29b-3p, miR-200b-3p, and miR-141-3p were significantly hypo-expressed after 12 h with respect to 30 min, whereas six miRNAs showed significant differences after 48 with respect to 12 h (miR-222-5p, miR-411-3p, miR-425-3p, hypo-expressed; miR200b-3p, miR-671-3p, miR-324-5p, hyper-expressed).

Overall, by considering also not significant (*p* > 0.05) expression values, most of microRNAs showed iso-expression after 30 min and hypo-expression after 12 and 48 h. However, in the 48 vs. 12 h comparison, expression level increase was observed in nearly half of miRNAs, possibly suggesting the induction of a putative feedback control mechanism to repristinate original conditions.

### 2.2. Identification of Target Genes and Pathways

To describe the putative functional role of the above-mentioned 21 differentially expressed miRNAs, an in silico analysis was performed by DIANA-miRpath v3.0 (https://dianalab.e-ce.uth.gr/html/mirpathv3/index.php?r=mirpath; accessed 6 July 2021). DIANA-TarBase and microT-CDS algorithms (genes union function) were used to identify experimentally supported and predicted miRNA/target-mRNA interactions, respectively, as well as significant KEGG pathways, in which target genes were involved (Table 2 and Table 3). Among them emerged the ECM-receptor interaction and the focal adhesion pathways, already described in cornea homeostasis and normal physiological functions [27], corneal epithelial cell motility [28], neovascularization [29] and adhesion [30,31], the Hippo signaling pathway, playing a role in limbal cell proliferation, corneal wound healing and regeneration [32,33] and, as expected, the Neurotrophin signaling pathway. The latter was considered for more in-depth analysis: sixty and fifty genes, targeted by 18 and 16 out of 21 miRNAs (miR-337-5p, mir-34a-3p, mir-1227-3p did not show any target gene) were detected by DIANA-TarBase and microT-based analysis, respectively (Table 4). Approximately 80 unique genes were identified by used algorithms, most of them targeted by more than one miRNA (Figure 2). Target genes, as represented in Figure 3, highlighted the involvement of major signaling pathways, such as MAPK, Ras, PI3K/Akt, p53, Jun, and NF-kB, known to induce cell growth, differentiation and survival, regulation of actin cytoskeleton, or apoptosis. Overall, miRNAs differentially expressed in response to rhNGF were able to regulate most of the genes specifically involved in significant signaling pathways included in the neurotrophin-induced molecular circuitries, playing a role in cell survival, proliferation, or, on the other hand, apoptosis.

### 2.3. Protein Expression Levels of Target AKT and RhoA

To perform an initial validation of in silico results, expression levels of total AKT and RhoA, two among the identified putative targets, were evaluated. As reported in Figure 2 and Table 4, *AKT1*, *AKT2*, and *AKT3* resulted in being target genes of miR-29b-3p, miR-27a-5p, and miR-324-5p (*AKT1*); miR-146a-5p and miR-29b-3p (*AKT2*); miR-30d-3p, miR-200b-3p, and miR-362-5p (*AKT3*). On the other hand, *RhoA* was reported to be regulated by miR-200b-3p and miR-27a-5p. Regarding *AKT*, as shown in Table 1, the above-mentioned miRs were mainly iso-expressed after 30 min and hypo-expressed after 12 and 48 h with respect to unstimulated cells. Moreover, expression levels of most of them increased after 48 when compared with those observed after 12 h (Table 1). As shown in Figure 4, total AKT expression globally reflects and is coherent with miRNAs’ expression levels, as they are comparable with the control after 30 min and hyper-expressed after 12 and, less markedly, 48 h. Overall, a similar and consistent RhoA protein expression, compared with levels of related miRNA, was observed in cells treated for 30 min and 12 h with rhNGF, followed by a decrease after 48 h (Figure 4). Even though it is known that circuitries involving miRNAs/target genes regulation are complex and can be also controlled by other different and multiple molecular interactions, this analysis could, however, be considered as an initial validation of the strength of the in silico results. Of note, RhoA/miR-200b-3p, AKT1/miR-29b-3p, AKT1/miR-27a-5p, AKT2/miR-29b-3p, and AKT2/miR-146a-5p interactions were experimentally supported in in vitro models by different methods (e.g., HITS-CLIP, microarray, luc assay, immunoprecipitation), as reported in TarBase and miRPathDB. The RhoA/miR-200b-3p interaction was further demonstrated in HEK293T cells [34].

## 3. Discussion

Human corneal epithelium expresses high-affinity TrkA receptors for NGF and releases growth factors (NGF, BDNF, glial cell derived neurotrophic factor GDNF, ciliary neurotrophic factor CNTF, epidermal growth factor EGF) that produce a positive effect on the survival, differentiation, and maturation of nerve fibers [10,35]. These molecules are the most important mediators of interactions between corneal epithelium and nerves, by activating reciprocally each other to secrete trophic neuropeptides and cytokines for the normal healing of the cornea [35]. For more than one decade, NGF has demonstrated its efficacy in maintaining corneal homeostasis in in vitro, ex vivo, and in vivo models and in preliminary uncontrolled studies on humans as well, where it was described as an effective treatment of corneal neurotrophic lesions [9,10,11,12,13]. NGF has now therapeutic value in ophthalmology, being approved as a first-in-line drug for neurotrophic keratitis (rhNGF, Cenegermin^®^). It is also under evaluation for other eye diseases of both anterior and posterior eye segments, such as dry eye disease (DED), glaucoma, and retinopathies [12,18,36]. However, very little is known about molecular mechanisms induced by NGF in corneal cells. In this study, we analyzed for the first time the modulation of microRNAs expression in response to rhNGF in an in vitro model of corneal epithelial cells. MicroRNAs are a class of short, non-coding RNA molecules crucial in the regulation, at the epigenetic level, of post-transcriptional gene expression. Their interaction with target genes is dynamic and varies depending on multiple factors, such as the relative quantity of miRNAs and related target mRNAs, affinity of their interactions, or subcellular location. Several miRNAs have been described in corneal physiology, disease, and potential therapy [37]. In this work, after analyzing a remarkable part of the human miRNome, we characterized a group of 21 miRNAs showing significant expression level variations after rhNGF treatment, most of which were never described in corneal cells. In this regard, the data in the literature related to cornea are available only for miR-146a-5p: in two studies [38,39], miR-146a hyper-expression was correlated with delayed corneal wound healing in telomerase-immortalized HCECs (human corneal epithelial cells) and LECs (limbal epithelial cells) models, most probably depending on the inhibition of activated p38 MAPK and EGFR, known as mediators of epithelial wound closure. MiR-146a was found hyper-expressed in patients with Sjogren’s syndrome, a condition characterized by dry eye, and in Sjogren-prone mice as well [40,41]. In addition, Movahedan et al. [42] described a *Notch1*-mediated decrease in early-stage corneal wound healing, and a recent study by using human primary limbal epithelial cells [43] demonstrated that miR-146a increased *Notch*1 by *Numb* downregulation. In accordance, both Targetscan human v7.2 (http://www.targetscan.org/vert_71/; accessed 6 October 2021) and miRDB (http://www.mirdb.org/; accessed 6 October 2021) databases report miR-146a-5p/*Numb* interaction, and previous studies on other cell types demonstrated the miR-146a-mediated *Numb* downregulation as well [44,45]. In this context, the NGF-induced miR-146a expression decrease detected in our in vitro model might lead to hypothesize a putative role of the neurotrophin in rebalancing corneal pathophysiological conditions by promoting, for instance, wound healing in the case of cornea damage. However, on the other hand, another study on a corneal alkali burn rat model showed that the treatment with miR-146a-overexpressing bone marrow mesenchymal stem cells induced a protective and repair effect by decreasing the levels of p65-NF-κB/PCNA (proliferating cell nuclear antigen) and by inhibiting apoptosis, inflammatory cytokine secretion and corneal neovascularization [46].

The twenty-one dysregulated miRNAs were further analyzed in silico by DIANA tools, and several significant pathways and target genes, predicted and/or experimentally validated, were identified. We focused our attention on the neurotrophin signaling pathway: approximately 80 unique target genes were identified, most of them targeted by more than one miRNA and playing a role in relevant signaling processes (i.e., *MAPK, PI3K/AKT, NF-kB*) and cell functions (e.g., differentiation, survival, or apoptosis) already described in corneal physiological and pathophysiological processes as being at the base of injury, wound healing, and regeneration [47,48,49,50]. The 80 genes represent the majority of those involved in the neurotrophin signaling pathway, as shown in the modified KEGG scheme (Figure 3), confirming the potential strong impact of miRNAs here identified on the modulation of specific NGF-induced cell responses. To date, few data about the NGF-mediated regulation of the above-mentioned target genes and related biological processes in corneal cells are available in literature. A study described epithelial corneal cells proliferation in response to cyclin D, PI3K/AKT, MAPK/Erk pathway activation (*AKT* and *MAPK* are target genes in our analysis) [51]; furthermore, in a corneal wound healing model of aquaporin5 knock-out mice, NGF was shown to induce epithelial and nerve regeneration by activating the Akt signaling pathway [52]. In vitro expansion of phenotypic and functional corneal endothelial cells able to contribute to restore corneal endothelium was described in a rabbit model in response to the combined activation of PI3K/Akt and Smad2 [53]. Park et al. [54] demonstrated NGF-induced attenuation of apoptosis and downregulation of Bax (here identified as target gene as well), NF-kB-p65, IL-1β, TNF-α, and cleaved-caspase 3 and reduction in inflammation in a diabetic high-glucose human corneal epithelial model. GTPase RhoA, Rac1, and Cdc42 proteins, identified as targets in this analysis, were described to maintain corneal functions and barrier homeostasis: Zhu et al. [55] observed expansion of HCECs monolayers due to RhoA-ROCK-non-canonical BMP (bone morphogenic protein)-NF-kB pathway activation, following p120 catenin and Kaiso proteins knocking-down; Cui et al. [56] demonstrated in vivo that the exposition to airborne particulate can induce delayed corneal epithelium wound closure by suppressing RhoA activity; Ortega et al. [57] showed that the activation of RhoA and Rac1 in response to stimulus disrupting endothelial cell–cell junctions preserved corneal endothelial barrier function; Rac1 was described as an important factor for corneal cell adhesion, motility promotion by fibronectin, and wound healing [58]; Cdc42 was able to promote wound repair and significantly improve the migration of human corneal epithelial cells in monolayer scratch assay [59]. Other genes here identified were the subject of previous studies: Rogge et al. [60] demonstrated the potential therapeutic efficacy of soluble Fas ligand (FASLG), which decreased inflammation and angiogenesis in primary and recurrent forms of herpetic stromal keratitis in a mouse model; a study based on a proteomic approach aimed at identifying putative biomarkers in tears from patients affected by DED secondary to rheumatoid arthritis highlighted reduced expression of SHC transforming isoform 1 (SHC1) in 63% of cases [61]. Taken together, the data in the literature so far available are overall consistent with the results obtained in this study: in fact, all of the above-mentioned genes are targeted by miRNAs that are hypo-expressed after NGF treatment, leaving one to hypothesize about the expression increase in protein products and evidencing a putative NGF-induced biological response on corneal cells in the presence of injury. Of note, due to the complexity of miRNA-mediated gene expression regulatory networks, in which multiple and different molecular mechanisms may be involved, we cannot exclude that other regulatory systems (e.g., DNA methylation or histone modification) or direct effects on genes/proteins involved in biochemical signaling may be implicated and contribute to rhNGF-induced response. However, although data here have shown a need further experimental validation, the analysis of AKT and RhoA protein expression in response to NGF treatments might be considered a preliminary demonstration of the strength of in silico results.

Overall, this research makes available interesting and novel insights in terms of NGF-regulated miRNAs and target genes identification, shedding light on some still unknown and possible biological mechanisms induced by NGF in corneal epithelial cells. The emerging role of microRNAs in putatively modulating and directing the response of corneal epithelial cells to NGF here evidenced is potentially of relevant impact. MicroRNAs are RNase-protected and very stable molecules, even under difficult conditions or extended storage. For this reason, they are considered to be suitable biomarkers for diagnosis, and if dysregulated, they are considered to be therapy targets to repristinate normal physiological conditions. Innovative therapeutic strategies are based on the use of synthetic miRNA inhibitors or mimics molecules, which, when appropriately delivered, can inhibit or restore normal expression levels and functions of hyper-expressed or downregulated miRs [62]. In this scenario, the characterization of miRNAs dysregulation in corneal biological processes, or in disease pathogenesis and progression, might be of translational value, this body part being easily accessible by targeted therapies, and the eye being one of the few organs where gene silencing could be successfully achieved. This innovative therapeutic approach is under evaluation for some eye disorders in several, mainly preclinical, trials. Simple local administration of naked siRNAs, a class of double stranded RNA fully complementary to mRNA and able to silence genes by endonucleolytic cleavage, demonstrated reduced systemic side effects and toxicity [63]. Besides the diseases that are high-impact and studied more in-depth, such as cancer or cardiovascular diseases [64,65,66,67], the possibility of using miRNAs for the treatment of anterior and/or posterior eye segment disorders is currently under investigation [68,69,70]. Arguably, this requires more exhaustive understanding of their specific role in this organ and related diseases. In this scenario, by improving knowledge in the field, miRNAs might represent in the future a suitable method for the treatment of ocular disorders.

In conclusion, this study analyzed the effects of rhNGF on epithelial corneal cells by focusing the attention on miRNAs expression levels, target genes, and pathways, and the study provided novel insights about molecules and possible responses induced by this neurotrophin in epithelial corneal tissue, potentially exploitable for clinical applications.

## 4. Materials and Methods

### 4.1. Cell Culture and Treatment

Human corneal epithelial cells (HCEpiC) (P10871, Innoprot, Derio, Bizkaia, Spain) were cultured in corneal epithelial cell medium (Innoprot, Derio, Bizkaia, Spain) supplemented with 5% fetal bovine serum (Gibco by Life Technologies Limited, Paisley, UK), 1% corneal epithelial cell growth supplement (CEpiCGS) (Innoprot), and 1% penicillin/streptomycin solution (Innoprot, Derio, Bizkaia, Spain). Cells were grown on T75 flasks precoated with collagen I (Corning^®^ BioCoat™ Collagen I Cultureware, Glendale, Arizona, USA) and maintained in humidified atmosphere at 37 °C with 5% CO_2_. The medium was changed every 3 days until the culture was approximately 70% confluent, and subsequently every day until the culture was approximately 80% confluent. Then, rhNGF (kindly provided by Dompé SpA, L’Aquila, Italy) at 200 ng/mL [71] was added to the medium. Afterwards, cells were incubated at 37 °C for three time points: 30 min and 12 h and 48 h. Untreated control cells were cultured as well, in parallel.

### 4.2. Total RNA Isolation

Total RNA, including the fraction less than 200 nucleotides in length, was isolated from HCEpiCs using the mirVana isolation kit (Invitrogen, Thermo Fisher Scientific, Waltham, MA, USA), according to the manufacturer’s protocol. The concentration of RNA was assessed by Nanodrop (Thermo Fisher Scientific, Waltham, MA, USA), and RNA samples were stored at −80 °C until use.

### 4.3. Reverse Transcription and TaqMan miRNA Array

Total RNAs from treated samples and control were reverse transcribed using the Megaplex RT primer Human Pool A and B set v3.0 (Applied Biosystem, Thermo Fisher Scientific, Waltham, MA, USA) and the TaqMan miRNA RT kit (Applied Biosystem, Thermo Fisher Scientific, Waltham, MA, USA). The reverse transcription reaction was performed by using 700 ng of total RNA, according to the manufacturer’s instructions. Subsequently, three replicated samples from each experimental time point and control cells were run on microfluidic TaqMan Array Human MicroRNA cards v3.0 (set A and B) (Applied Biosystems), according to the manufacturer’s instructions. These 384-well cards are preconfigured for a total of 754 unique assays specific to human miRNAs. Samples were analyzed on a ViiA7 instrument (Applied Biosystems, Thermo Fisher Scientific, Waltham, MA, USA), and data were processed by QuantStudio Real-Time PCR software v1.2 (Applied Biosystems, Thermo Fisher Scientific, Waltham, MA, USA). MicroRNA’s expression levels (RQ, relative quantification) were evaluated by comparative assay (2^−∆∆Ct^) [72]. snU6 RNA was used as endogenous control.

### 4.4. Statistics and Target Genes/Pathways Analysis

Comparative data analysis was carried out by ExpressionSuite v1.1 (Thermo Fisher, Waltham, MA, USA). To make the study as stringent as possible, the confidence interval was set as >95%, and wells with flag(s) (abnormal/doubt signals and/or parameters) were omitted from the analysis. Global distribution of microRNAs’ expression was visualized by volcano plots. Among the differentially expressed (RQ, fold change ≤0.5 or ≥2), a list of significant miRs (*p* < 0.05) was generated by using the proprietary algorithm (Thermo Fisher, Waltham, MA, USA) based on ∆Ct. Significant miRNAs were further analyzed by DIANA tools (DIANA-miRpath v.3, -microT-CDS,-TarBase v7.0) [73] to identify predicted (microT-CDS database) [74] and/or experimentally supported (TarBase database) [75] target genes and pathways. We used KEGG database annotation with standard statistics and genes union settings for an exploratory functional analysis. MiRNAs and target genes were included in a Neo4J graph database [76] and elaborated to represent relationships. qRT-PCR data files are presented in Appendix A.

### 4.5. Western Blot Analysis

To assess the efficacy of rhNGF treatment under our cell system conditions, immunoblotting was performed to investigate the expression of phosphorylated ERK1 and ERK2 MAP kinases.

HCEpiCs were lysed in 50 μL modified RIPA Buffer (PBS 1X, NP40 1%, sodium deoxycholate 0.5%, SDS 1%, Complete-Mini protease inhibitor cocktail tablet (Roche Diagnostics, Basel, Switzerland), PMSF 10 mg/mL, aprotinin 10 mg/mL, and sodium orthovanadate 0.1 M (Sigma-Aldrich, Burlington, MA, USA). Cell lysates were incubated on ice for 30 min and then centrifuged at 14,000× *g* for 15 min at 4 °C. The supernatant was recovered and stored at −80 °C until use.

Protein concentration was determined using the standard BCA control with the Pierce BCA protein assay kit, according to the manufacturer’s instructions.

A total of 30 μg of protein extracts was loaded onto an SDS-PAGE and subjected to electrophoresis; then proteins were electro-transferred to a 0.2 μm nitrocellulose membrane (Amersham™ Protran™ Premium, Ge Healthcare, Little Chalfont, Buckinghamshire, UK) and hybridized overnight at 4 °C with the phospho-p44/42 MAPK (ERK1/2) monoclonal antibody (# 4370, Cell Signaling Technology, Danvers, MA, USA). The membrane was washed in TBS-T and incubated one hour at room temperature with the goat anti-rabbit IgG-HRP secondary antibody (sc-2030, Santa Cruz Biotechnology, Dallas, Texas, USA). After signal detection, the membrane was stripped (Restore™ Western Blot Stripping Buffer, Thermo Fisher Scientific, Waltham, MA, USA) for 30 min at 37 °C, then it was incubated again overnight at 4 °C with the p44/42 MAPK (ERK1/2) monoclonal antibody (#4695, Cell Signaling Technology, Danvers, MA, USA) and with secondary antibody as previously described. Actin (sc-1615, Santa Cruz Biotechnology, Dallas, Texas, USA) was used as endogenous control. The same procedure was followed with anti-AKT antibody (#9272, Cell Signaling Technology, Dallas, Texas, USA), recognizing AKT1, 2 and 3 isoforms, and anti-RhoA (#2117, Cell Signaling Technology, Dallas, Texas, USA); chemiluminescent detection system (SuperSignal™ West Pico Plus, Thermo Scientific, Waltham, MA, USA) and ChemiDoc XR + (Bio-Rad, Hercules, California, USA) system or autoradiography films were used to detect signals. Bands corresponding to proteins of interest were scanned and quantified by densitometry using ImageJ software (https://imagej.nih.gov/ij/ accessed 6 October 2021).

## Figures and Tables

**Figure 1 ijms-23-03597-f001:**
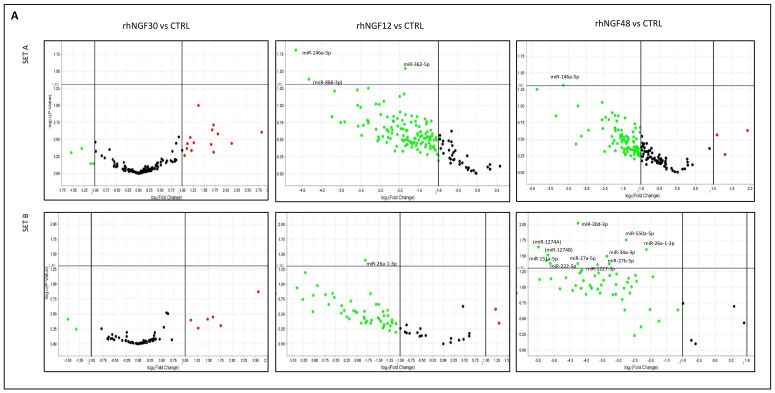
MiRNA expression in HCEpiCs in response to rhNGF. Volcano plots from human microRNA array cards Set (**A**) (top) and Set (**B**) (bottom) analysis, representing levels of miRNAs with respect to unstimulated cells (CTRL) (**A**) and, dynamically, through experimental time points (**B**). Significant miRNAs IDs are reported close to the corresponding plot. *x* axis: fold change (RQ, relative quantification, log scale); *y* axis: *p*-value (log scale). Horizontal blue line: *p* = 0.05 threshold. Between brackets: miRs not currently listed in the MiRbase database.

**Figure 2 ijms-23-03597-f002:**
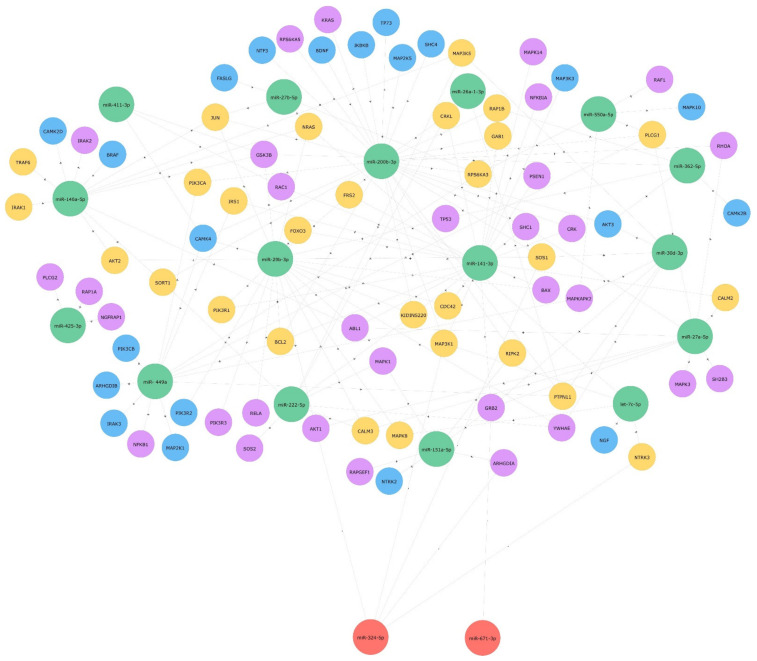
MiRNA-target genes interactions in neurotrophin signaling pathway. Neo4J graph of relationships between significant miRNAs and target genes. Colors of nodes as follows: green: hypo-expressed miRNAs; red: hyper-expressed miRNAs; blue: predicted target genes from DIANA-microT analysis; violet: experimentally supported target genes from DIANA-TarBase analysis; yellow: target genes identified by both DIANA-microT and TarBase analysis. Symbols as follows: +: induction of target gene upregulation; −: induction of target gene downregulation.

**Figure 3 ijms-23-03597-f003:**
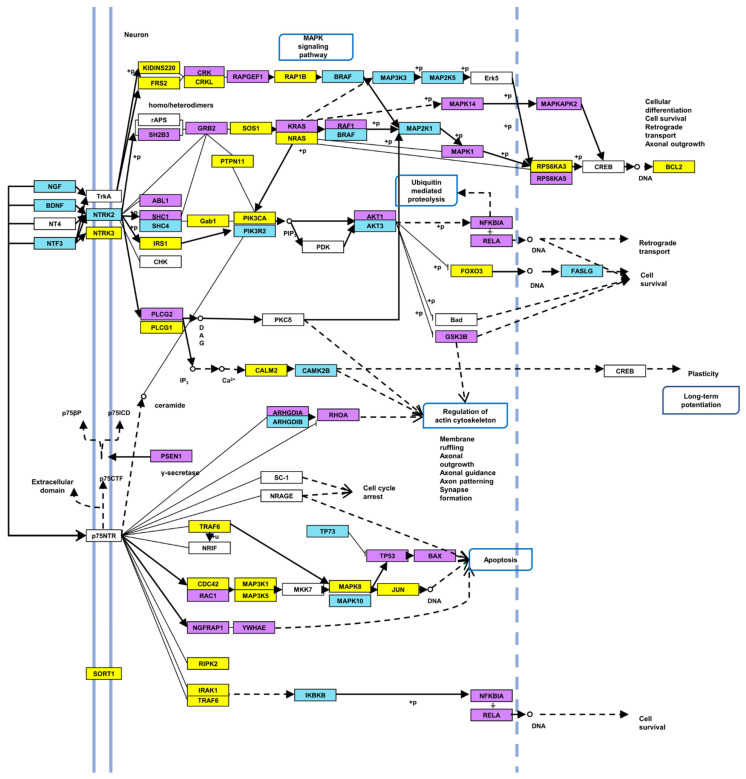
Target genes of significant miRNAs in neurotrophin signaling pathway. Colors of boxes as follows: blue: predicted target genes from DIANA-microT analysis; violet: experimentally-supported target genes from DIANA-TarBase analysis; yellow: target genes identified by both DIANA-microT and TarBase algorithms; white: genes not identified as target of significant miRs here described.Arrows as follows: regular line: molecular interaction or relation; dotted line: indirect link or unknown reaction. Modified from DIANA tools/KEGG.

**Figure 4 ijms-23-03597-f004:**
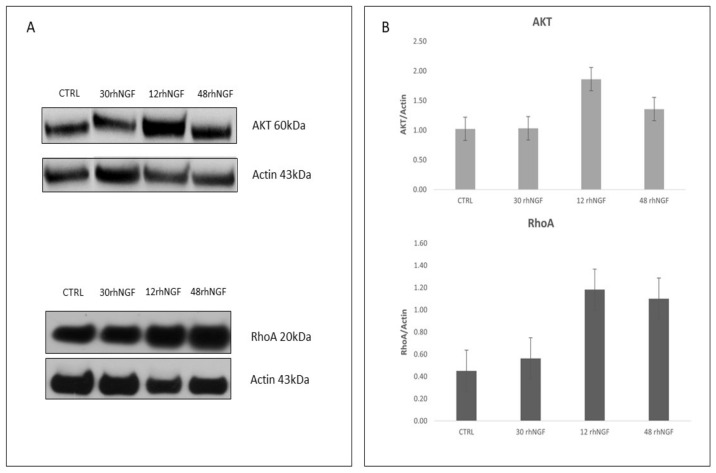
Protein expression levels of target AKT and RhoA. (**A**) Western blot analysis of protein extracts from cells treated with rhNGF for the reported time points. (**B**) Densitometric analysis of immunoblotting shown in A. Data are mean ± SE of two independent experiments for each target.

**Table 1 ijms-23-03597-t001:** Dynamic expression of miRNAs significantly dysregulated (n = 21) at least in 1 comparison, after rhNGF treatment. Relative quantification (RQ) data obtained by comparing 30 min and 12 and 48 h NGF-treated vs. untreated cells (NGF30, NGF12, NGF48), NGF 12 h vs. 30 min (NGF12vs30) and NGF 48 vs. 12 h (NGF 48vs12). Green: RQ values expressing significant downregulation, red: RQ values expressing significant upregulation. Significant *p*-values in bold.

miRNA miRbase ID	NGF30	*p*-Value	NGF12	*p*-Value	NGF48	*p*-Value	NGF12vs30	*p*-Value	NGF48vs12	*p*-Value
hsa-let7c-5p	7.035	0.248	1.491	0.773	2.485	0.538	** 0.212 **	**0.003**	1.629	0.330
hsa-mir-29b-3p	4.379	0.362	0.5	0.645	0.955	0.977	** 0.114 **	**0.038**	1.917	0.536
hsa-miR-449a	3.279	0.194	1.199	0.797	1.85	0.436	** 0.368 **	**0.000**	1.548	0.255
hsa-miR-337-5p	3.215	0.229	0.661	0.617	2.105	0.980	** 0.207 **	**0.016**	3.414	0.980
hsa-miR-671-3p	1.819	0.634	0.399	0.374	3.819	0.234	0.219	0.207	** 9.616 **	**0.009**
hsa-mir-1227-3p	1.292	0.823	0.44	0.397	** 0.08 **	**0.044**	0.343	0.361	0.179	0.088
hsa-miR-26a-1-3p	1.264	0.724	** 0.285 **	**0.04**	** 0.228 **	**0.025**	0.226	0.080	0.799	0.560
hsa-miR-27b-5p	1.175	0.870	0.34	0.218	** 0.102 **	**0.043**	0.291	0.225	0.303	0.071
hsa-miR-141-3p	1.175	0.867	0.118	0.059	0.514	0.578	** 0.101 **	**0.048**	4.192	0.306
hsa-miR-200b-3p	1.139	0.903	0.125	0.106	0.373	0.340	** 0.109 **	**0.047**	** 2.936 **	**0.049**
hsa-mir-425-3p	1.043	0.973	0.661	0.649	0.104	0.065	0.631	0.687	** 0.155 **	**0.023**
hsa-mir-550a-5p	1.038	0.961	0.311	0.144	** 0.147 **	**0.018**	0.301	0.211	0.471	0.296
hsa-mir-222-5p	1.013	0.993	0.388	0.482	** 0.029 **	**0.042**	0.384	0.523	** 0.073 **	**0.046**
hsa-mir-34a-3p	0.974	0.976	0.168	0.143	** 0.097 **	**0.032**	0.173	0.158	0.572	0.578
hsa-miR-146a-5p	0.973	0.976	** 0.039 **	**0.015**	** 0.114 **	**0.049**	** 0.04 **	**0.013**	2.854	0.125
hsa-mir-151a-5p	0.968	0.984	0.236	0.278	** 0.027 **	**0.038**	0.244	0.361	0.111	0.057
hsa-mir-27a-5p	0.863	0.909	0.226	0.193	** 0.052 **	**0.042**	0.261	0.289	0.23	0.105
hsa-mir-411-3p	0.725	0.980	1.391	0.840	0.203	0.430	1.927	0.980	** 0.184 **	**0.015**
hsa-miR-324-5p	0.712	0.722	0.173	0.097	0.446	0.186	0.245	0.221	** 2.639 **	**0.045**
hsa-miR-30d-3p	0.682	0.682	0.194	0.304	** 0.052 **	**0.01**	0.285	0.446	0.268	0.394
hsa-miR-362-5p	0.504	0.347	** 0.276 **	**0.029**	0.271	0.160	0.548	0.418	0.939	0.940

**Table 2 ijms-23-03597-t002:** DIANA-TarBase pathway analysis in response to rhNGF. KEGG pathways identified by considering significant dysregulated miRNAs (n = 21), as detected by DIANA-TarBase (experimentally supported) analysis. The pathway analyzed in this study is highlighted in red.

KEGG Pathway	*p*-Value	#genes	#miRNAs
ECM-receptor interaction	1.94 × 10^−31^	39	15
Adherens junction	1.37 × 10^−11^	47	16
Proteoglycans in cancer	1.75 × 10^−9^	85	19
Prion diseases	3.82 × 10^−9^	12	15
Viral carcinogenesis	5.99 × 10^−9^	91	17
Focal adhesion	1.90 × 10^−7^	102	18
Protein processing in endoplasmic reticulum	2.99 × 10^−7^	87	18
Pathways in cancer	1.04 × 10^−6^	163	19
Fatty acid biosynthesis	1.76 × 10^−6^	4	5
Chronic myeloid leukemia	3.63 × 10^−6^	42	17
Cell cycle	5.44 × 10^−6^	62	17
Glioma	5.44 × 10^−6^	34	18
Renal cell carcinoma	5.44 × 10^−6^	38	19
Hepatitis B	5.79 × 10^−6^	66	17
Oocyte meiosis	6.01 × 10^−6^	55	18
Bacterial invasion of epithelial cells	6.63 × 10^−6^	41	16
Ubiquitin mediated proteolysis	7.71 × 10^−6^	70	17
Prostate cancer	1.30 × 10^−5^	47	17
Neurotrophin signaling pathway	2.11 × 10^−5^	60	18
Small cell lung cancer	3.09 × 10^−5^	46	17
PI3K-Akt signaling pathway	4.53 × 10^−5^	141	18
Transcriptional misregulation in cancer	7.92 × 10^−5^	70	19
Hippo signaling pathway	9.95 × 10^−5^	59	18
Central carbon metabolism in cancer	0.000127	34	16
p53 signaling pathway	0.000134	37	17
Shigellosis	0.000227	33	14
Colorectal cancer	0.000227	33	16
Lysine degradation	0.000344	20	14
FoxO signaling pathway	0.000344	62	17
Endocytosis	0.000446	84	17
Acute myeloid leukemia	0.000496	29	17
Endometrial cancer	0.000515	27	16
Pancreatic cancer	0.000515	34	17
TGF-beta signaling pathway	0.000952	36	17
Epstein–Barr virus infection	0.001042	87	17
Sulfur metabolism	0.004825	5	5
Fatty acid elongation	0.005702	6	6
Non-small cell lung cancer	0.005702	27	17
Bladder cancer	0.005702	22	18
NF-kappa B signaling pathway	0.006256	32	17
Arrhythmogenic right ventricular cardiomyopathy (ARVC)	0.006413	26	16
HIF-1 signaling pathway	0.006698	47	18
Spliceosome	0.007512	56	18
Amoebiasis	0.011039	41	14
Sphingolipid signaling pathway	0.011162	49	18
Regulation of actin cytoskeleton	0.019688	79	16
AMPK signaling pathway	0.019688	53	17
Thyroid hormone signaling pathway	0.019688	52	19
Circadian rhythm	0.023121	16	13
mRNA surveillance pathway	0.024669	40	16
RNA transport	0.024669	64	18
Pathogenic Escherichia coli infection	0.024739	26	13
Dorso-ventral axis formation	0.024739	15	15
Insulin signaling pathway	0.024739	57	16
Hepatitis C	0.024739	53	17
N-Glycan biosynthesis	0.037600	19	14
Prolactin signaling pathway	0.039423	32	17
Thyroid cancer	0.048754	14	16
Melanoma	0.049562	29	17

**Table 3 ijms-23-03597-t003:** DIANA-microT-CDS pathway analysis in response to rhNGF. KEGG pathways identified by considering significant dysregulated miRNAs (n = 21), as detected by DIANA-microT-CDS (predicted) analysis. The pathway analyzed in this study is highlighted in red.

KEGG Pathway	*p*-Value	#genes	#miRNAs
ECM-receptor interaction	1.74 × 10^−15^	34	15
Prion diseases	1.12 × 10^−8^	9	9
ErbB signaling pathway	1.12 × 10^−8^	46	13
Glioma	3.13 × 10^−6^	32	13
Focal adhesion	3.13 × 10^−6^	92	16
Proteoglycans in cancer	3.13 × 10^−6^	84	19
Renal cell carcinoma	3.35 × 10^−6^	36	12
Glycosaminoglycan biosynthesis—heparan sulfate/heparin	5.01 × 10^−6^	14	8
Choline metabolism in cancer	1.32 × 10^−5^	51	14
FoxO signaling pathway	0.000104	58	14
PI3K-Akt signaling pathway	0.000104	127	20
Amoebiasis	0.000116	43	14
Adherens junction	0.000182	37	14
Lysine degradation	0.000186	19	10
mTOR signaling pathway	0.000192	33	14
Thyroid hormone signaling pathway	0.000211	49	18
Ras signaling pathway	0.000345	85	17
Rap1 signaling pathway	0.000661	84	16
Axon guidance	0.000768	51	15
Glycosaminoglycan biosynthesis—keratan sulfate	0.001064	8	6
Pathways in cancer	0.001065	142	18
Adrenergic signaling in cardiomyocytes	0.001896	54	18
p53 signaling pathway	0.004244	31	14
TGF-beta signaling pathway	0.004617	33	16
Glycosaminoglycan biosynthesis—chondroitin sulfate/dermatan sulfate	0.004672	8	6
Small cell lung cancer	0.005864	38	12
Neurotrophin signaling pathway	0.005864	50	16
Hippo signaling pathway	0.006055	49	15
HIF-1 signaling pathway	0.006421	45	14
Glycosphingolipid biosynthesis—lacto and neolacto series	0.006829	11	8
AMPK signaling pathway	0.007480	49	18
Prostate cancer	0.008563	38	13
Prolactin signaling pathway	0.008642	27	12
Circadian rhythm	0.008642	17	15
cGMP-PKG signaling pathway	0.100938	62	17
Regulation of actin cytoskeleton	0.012869	76	15
Phosphatidylinositol signaling system	0.014451	32	13
Biotin metabolism	0.014567	1	1
Long-term depression	0.014567	26	13
Sphingolipid signaling pathway	0.014740	44	16
Non-small cell lung cancer	0.019923	24	10
MAPK signaling pathway	0.021152	90	17
Pancreatic cancer	0.024846	26	10
Melanoma	0.024846	30	12
Estrogen signaling pathway	0.029488	35	15
Wnt signaling pathway	0.035309	55	16
Chagas disease (American trypanosomiasis)	0.038162	38	15
Bacterial invasion of epithelial cells	0.042141	28	13
Adipocytokine signaling pathway	0.043705	27	13

**Table 4 ijms-23-03597-t004:** MiRNAs and target genes in neurotrophin signaling pathway, as detected by DIANA-TarBase (experimentally supported, in violet), microT (predicted, in light blue), and both (yellow). As shown, several genes were detected by both algorithms. MiRs displaying global down- or upregulation after NGF treatment are represented in green and red, respectively.

miR-222-5p	miR-151a-5p	miR-29b-3p	miR-26a-1-3p	miR-30d-3p	miR- 449a	miR-550a-5p	miR-141-3p	miR-27a-5p	miR-200b-3p	miR-146a-5p	miR-324-5p	miR-425-3p	miR-362-5p	miR-411-3p	miR-27b-5p	miR-671-3p	let-7c-5p
** CALM3 **	** ARHGDIA **	** ABL1 **	** PTPN11 **	** AKT3 **	** ARHGDIB **	** MAPK10 **	** ABL1 **	** ABL1 **	** AKT3 **	** ABL1 **	** AKT1 **	** NGFRAP1 **	** AKT3 **	** CAMK4 **	** FASLG **	** GRB2 **	** MAP3K1 **
** CRKL **	** GRB2 **	** AKT1 **		** GRB2 **	** BCL2 **	** MAPKAPK2 **	** BAX **	** AKT1 **	** BCL2 **	** AKT2 **	** ARHGDIA **	** PLCG2 **	** CAMK2B **	** GSK3B **	** JUN **		** MAPK8 **
** KIDINS220 **	** MAPK1 **	** AKT2 **		** MAP3K5 **	** CAMK4 **	** PLCG1 **	** BCL2 **	** CALM2 **	** BDNF **	** BRAF **	** MAP3K1 **	** RAP1A **	** GSK3B **	** RAC1 **	** NRAS **		** NGF **
** MAP3K5 **	** NTRK2 **	** BAX **		** MAPKAPK2 **	** IRAK3 **	** RAF1 **	** CALM2 **	** CALM3 **	** CAMK4 **	** CAMK2D **	** NTRK3 **		** RAP1B **		** RPS6KA3 **		** NRAS **
** MAPK1 **	** RAPGEF1 **	** BCL2 **		** PTPN11 **	** IRS1 **	** RPS6KA3 **	** CDC42 **	** GRB2 **	** CRK **	** GSK3B **	** SOS1 **		** TP53 **				** NTRK3 **
** SORT1 **	** RIPK2 **	** CALM3 **		** SOS1 **	** MAP2K1 **		** CRK **	** MAPK3 **	** CRKL **	** IRAK1 **							
** SOS2 **		** CAMK4 **		** YWHAE **	** MAPKAPK2 **		** CRKL **	** PLCG1 **	** FRS2 **	** IRAK2 **							
** YWHAE **		** CDC42 **			** NFKB1 **		** FOXO3 **	** RHOA **	** GAB1 **	** JUN **							
		** FOXO3 **			** PIK3CA **		** FRS2 **	** SH2B3 **	** IKBKB **	** MAP3K5 **							
		** FRS2 **			** PIK3CB **		** GAB1 **	** SHC1 **	** JUN **	** NRAS **							
		** GSK3B **			** PLCG1 **		** GRB2 **		** KIDINS220 **	** SORT1 **							
		** JUN **			** PTPN11 **		** IRS1 **		** KRAS **	** TRAF6 **							
		** KIDINS220 **			** RPS6KA3 **		** KIDINS220 **		** MAP2K5 **								
		** MAPK8 **					** MAP3K1 **		** MAP3K1 **								
		** NRAS **					** MAP3K3 **		** NTF3 **								
		** PIK3R1 **					** MAPK14 **		** PIK3CA **								
		** PIK3R2 **					** NFKBIA **		** PLCG1 **								
		** PIK3R3 **					** PIK3R1 **		** PTPN11 **								
		** RELA **					** PSEN1 **		** RAP1B **								
		** SOS1 **					** RAC1 **		** RHOA **								
		** TP53 **					** SHC1 **		** RIPK2 **								
		** YWHAE **							** RPS6KA3 **								
									** RPS6KA5 **								
									** SHC1 **								
									** SHC4 **								
									** SORT1 **								
									** SOS1 **								
									** TP73 **								

## Data Availability

The original contributions presented in the study are included in the article. Further inquiries can be directed to the corresponding author.

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
