# Peer review of "MicroRNAs Expression in Response to rhNGF in Epithelial Corneal Cells: Focus on Neurotrophin Signaling Pathway"

_ijms, 2022, doi:10.3390/ijms23073597_

Round 1
Reviewer 1 Report
The paper from Chiara Compagnoni et al., describes miRNAs expression in HCEpiC after time-dependent rhNGF treatment. In this study, they found that twenty-one miRNAs were significantly regulated in response to rhNGF. However, there are some critical concerns regarding their designed study.
- HCEpiC were treated using rhNGF for 30 minutes, 12 hours, and 48 hours. Please clarify why these time points were chosen. .
- In the “1 MiRNA expression in HCEpiC cells in response to rhNGF”, of “Results”, the authors provide different miRNAs. Please provide the most highly expressed miRNA list to help the reader obtain a comprehensive understanding of the effects of rhNGF on miRNA expression levels in HCEpiC.
- In the “4.5. Western blot analysis” of “Materials and Methods”, the authors described the following:“protein concentration was determined using BCA...A total of 30 μg of protein extracts were loaded onto a SDS-PAGE…” (lines 386-388). So the levels of endogenous control β-Actin are substantially in agreement. If not, β-Actin may be not an appropriate endogenous control. From Figure 4 (lines208), the results are unacceptable. Lack of consistency suggests that the experimental skills may be problematic.
- Please combine the most highly expressed miRNA and the most highly differentially expressed miRNA to screen for more pertinent miRNAs. Then to confirm a link between AKT or RhoA and miRNAs, please provide luciferase activity assay.
- In this study, the authors think that rhNGF-induced all of its effects through miRNAs/targets. Nevertheless, there are other mechanisms that can mediate such control., For example,1) a direct effect of rhNGF on the expression of AKT ;2) or on RhoA; 3) or rhNGF effect on the expression of AKT; 4) or RhoA by another epigenetic regulation, for example DNA methylation and histone modification. Further description of these alternatives should be summarized in the Discussion.
- More important, in this study, the authors only used HCEC lines, and they did not design any animal experiments. Of course, their results would be more relevant if they can provide any confirmatory results with primary HCEpiC,

Author Response
Please, see the attachment

Reviewer 2 Report
Although nerve growth factor (NGF) is used in the treatment of several eye afflictions, little is known about the molecular mechanism by which NGF promotes healing. In this study, changes in miRNA expression in response to treatment of corneal cells with NGF were examined. Although the choice of a TaqMan array was a bit unusual, given the ubiquity and greater depth of RNA-seq, the authors were still able to find a number of miRNAs that were differentially expressed in the presence of NGF. They identified previously validated and predicted targets of these miRNAs and discovered that many were associated with the neurotrophin signaling pathway. Although lacking in experimental follow-up, this study offers a unique analysis of the miRNA response to NGF and lays the foundation for future research in this area.
Issues
- The main weakness of this paper is the lack of experimental evidence to confirm the relationships between the dysregulated miRNAs discovered in the array and the target genes identified in silico. The array and Western blots provide good circumstantial evidence, but more work is needed. For example, a more specific interaction could be shown by inhibiting the miRNAs that target AKT and RhoA in the absence of rhNGF in order to see if these genes are upregulated.
- The type in figures 2 and 3 is too small to read.
Round 2
Reviewer 2 Report
Accept with minor grammar correction.
Author Response
The authors thank the Reviewer for suggestions to improve the quality of the manuscript.
We carefully checked for grammar corrections and we hope to satisfactorily have addressed the requests.